# Identity and Maturity of iPSC-Derived Oligodendrocytes in 2D and Organoid Systems

**DOI:** 10.3390/cells13080674

**Published:** 2024-04-13

**Authors:** Ella Zeldich, Sandeep Rajkumar

**Affiliations:** 1Department of Anatomy & Neurobiology, Boston University Chobanian and Avedesian School of Medicine, Boston, MA 02118, USA; 2Center for Systems Neuroscience, Boston University, Boston, MA 02115, USA; 3Neurophotonics Center, Boston University, Boston, MA 02115, USA

**Keywords:** iPSCs, brain organoids, oligodendrocytes, myelin, regional patterning

## Abstract

Oligodendrocytes originating in the brain and spinal cord as well as in the ventral and dorsal domains of the neural tube are transcriptomically and functionally distinct. These distinctions are also reflected in the ultrastructure of the produced myelin, and the susceptibility to myelin-related disorders, which highlights the significance of the choice of patterning protocols in the differentiation of induced pluripotent stem cells (iPSCs) into oligodendrocytes. Thus, our first goal was to survey the different approaches applied to the generation of iPSC-derived oligodendrocytes in 2D culture and in organoids, as well as reflect on how these approaches pertain to the regional and spatial fate of the generated oligodendrocyte progenitors and myelinating oligodendrocytes. This knowledge is increasingly important to disease modeling and future therapeutic strategies. Our second goal was to recap the recent advances in the development of oligodendrocyte-enriched organoids, as we explore their relevance to a regional specification alongside their duration, complexity, and maturation stages of oligodendrocytes and myelin biology. Finally, we discuss the shortcomings of the existing protocols and potential future explorations.

## 1. Introduction

Oligodendrocytes, the myelinating cells of the central nervous system (CNS), are chiefly responsible for myelin ensheathment of axons, providing trophic support and protection to neurons [1,2,3,4], and regulating iron homeostasis [3,5,6]. In the spinal cord, oligodendrocytes are produced by neuroepithelial (NE) cells in the ventral portion of the neural tube and are dependent on the proximity to the source of the morphogen sonic hedgehog (SHH) [7,8]. These ventrally-derived neural progenitor cells committing to oligodendrocyte lineage express basic helix-loop-helix (bHLH) oligodendrocyte transcription factor 2 (OLIG2) and NK2 Homeobox 2 (NKX2.2), followed by the expression of SRY-box transcription factor 10 (SOX10), which is consistent with oligodendrocyte fate commitment [9]. Both *OLIG2* and *NKX2.2* are considered to be the downstream targets of the SHH pathway [10,11]. While *OLIG2* and *NKX2.2* are mutually exclusive initially [12], their subsequent co-expression is essential for the differentiation of oligodendrocyte progenitor cells (OPCs) from NPCs [10,12,13]. Loss of functions of OLIG2 completely abolishes oligodendrocyte lineage commitment and development [14,15] and the expression of NKX2.2 is essential for the maturation of oligodendrocytes including the induction of myelin basic protein (MBP), and terminal oligodendrocyte differentiation [13,16,17]. Ventrally derived OPCs are characterized by the expression of platelet-derived growth factor receptor alpha (PDGFRα) and membrane proteoglycan NG2 migrate towards the dorsal portion of the neural tube. There is an additional population of oligodendrocytes driven by *OLIG2* expression that is produced in the dorsal portion of the neural tube independent of SHH [7,18,19,20]. The spatial origin across the dorsal-ventral axis determines cell density and functional properties of oligodendrocytes, as there are twice as many myelinating oligodendrocytes in the ventral axon tracts of the spinal cord compared to the dorsal tracts. There are also measurable differences in the length of myelin sheaths produced by oligodendrocytes with different regional cell identities [21,22].

In the forebrain, the first two waves of OPC genesis originate prenatally in the medial and lateral ganglionic eminence, respectively, followed by a third dorsally-derived wave originating in the postnatal cortex [23]. This ventro-dorsal sequential production of OPCs is present in the human brain as well [24,25,26]. It has been shown that there is a degree of functional redundancy as dorsally-derived oligodendrocytes can compensate for the loss of the ventrally-derived oligodendrocytes and adequately myelinate neuronal axons in the areas where ventrally-derived cells were eliminated [23]. However, recent studies suggest that distinct developmental sources contribute to OPC heterogeneity, population expansion [27], response to injury, and demyelination-related conditions [26].

Additional heterogeneity stems from a different spatial origin along the rostral–caudal axis. Spinal cord-derived OPCs exhibit distinct morphology (cell area and branching) and developmental trajectory compared to brain-derived OPCs [28]. The differences between the spinal cord OPCs and brain-derived OPCs are also reflected at the functional level as the regenerative capacity of OPCs, and the length of myelin sheaths produced by the spinal cord oligodendrocytes differ significantly from those of the forebrain-derived oligodendrocytes [22].

Induced pluripotent stem cell (iPSC)-derived modeling provides an indispensable platform to expand and deepen our knowledge of the molecular and functional biology of human CNS cells including neurons, astrocytes, and oligodendrocytes. Despite extensive technological development in the field achieved in recent years, modeling oligodendrocyte development in two-dimensional (2D) and brain organoid systems still presents a challenge. Some of the difficulties arise from the complexity of the events and steps in the commitment of neural progenitor cells to oligodendrocyte lineage, exposure to combinatorial control of the morphogens, and tight temporal and spatial regulation. While numerous recently established protocols enable the generation of iPSC-derived oligodendrocytes, few of them address the developmental and spatial origin, and the subsequent effects on the molecular, functional, and cellular characteristics of the generated cells. This regional heterogeneity becomes increasingly important in disease modeling. The suitability of the regional patterning of iPSC-derived cells should be carefully adjusted with respect to the investigated system or disease. However, oftentimes these aspects are not addressed, and brain-related disorders involving perturbed oligodendrocyte/myelin biology can end up being modeled with oligodendrocytes generated, for example, by exposure to retinoic acid (RA), thereby assigning the oligodendrocytes with a spinal-cord-like fate.

In this review, we present the recent approaches for the generation of iPSC-derived oligodendrocytes in 2D and 3D, with an emphasis on organoid systems. We cover a variety of approaches to produce human iPSC-derived oligodendrocytes, including the readouts related to the appearance of oligodendrocyte markers across the development, detection of myelin ensheathment, and duration of the protocols. We shed light on the application of morphogens and small molecules utilized for the regional patterning of oligodendrocytes. We also aim to highlight the current limitations in targeted oligodendrocyte differentiation and myelination-related aspects.

## 2. Oligodendrocytes and Myelin in iPSC-Derived 2D Models

Extensive studies targeting neurodevelopmental processes related to the oligodendrocyte lineage commitment and differentiation in vivo paved the way for the establishment of in vitro protocols. This enabled the generation of the first human oligodendrocytes from human embryonic stem cells (ESCs) and subsequently from human iPSCs.

One of the key steps of the protocols was the definition of the cells committing to oligodendrocyte fate. As such, the co-expression of OLIG2 and NKX2.2, in neuroepithelial cell/neural progenitor cells (NECs/NPCs), was defined as essential for the commitment of NPCs to glial lineage, and the subsequent expression of the transcription factor SOX10, marking oligodendrocyte lineage [16,29,30]. Revel’s group [16] showed that *NKX2.2* can be induced by the treatment of the ESCs with the caudalizing factor RA [31]. They also showed that the inhibition of the BMP pathway through the application of noggin treatment is necessary for the development of mature, ramified oligodendrocytes expressing MBP [16]. As the secretion of BMPs originates from the roof plate and contributes to the generation of the dorsoventral gradient through diffusion [32,33], inhibition of the MBP pathway potentially promotes the ventral patterning of NPCs and their differentiation into pre-OPCs and early OPCs. The activation of the proteins in the BMP signaling pathway mediates the inhibitory effect on the oligodendrocyte population [34]. These principles lay the foundation for the initial protocols leading to the generation of mature human oligodendrocytes derived from ESCs and iPSCs that are capable of active myelination in vitro and in vivo.

In a study conducted by Zhang’s group [35], human ESCs were differentiated into PAX6 and SOX1 positive NECs/NPCs through the generation of embryoid bodies (EBs) as aggregates in suspension. The ventral and caudal patterning was achieved via the treatment with morphogen sonic hedgehog (SHH) and RA, respectively. Their differentiation into motor neurons, physiologically originating from the same embryonic domains, was prevented by the removal of RA. The application of fibroblast growth factor 2 (FGF2) served to produce a mitogenic effect on neural progenitors, thus increasing the pool of the pre-OPCs [7,35]. The generation of the pre-OPCs was deemed successful if the cells co-expressed the transcription factors OLIG2 and NKX2.2. However, to allow the transition from pre-OPCs to OPCs, FGF2 was omitted from the media composition as it inhibits SHH signaling and thus represses OPC specification [7]. The subsequent differentiation of the pre-OPCs into OPCs was achieved through the application of a so-called “glial medium” containing factors that promote the survival and proliferation of OPCs, such as platelet-derived growth factor-AA (PDGF-AA), insulin-like growth factor 1 (IGF1), a cyclic AMP (cAMP) analog, biotin, and triiodo-L-thyronine (T3). This 90-day protocol successfully yielded OPCs; however, no attempt was made to induce their further maturation or to characterize the regional/spatial properties of the generated OPCs. Exposure to RA as a part of the differentiation protocol leads most likely to the “spinal cord”-like OPCs that differ from the brain-like OPCs both functionally and transcriptomically [22,36,37].

Based on these and the previously published works, three important technical and conceptual aspects were incorporated into the subsequently developed protocols for the generation of oligodendrocytes. First, the utilization of morphogens and small molecules drives the ventral-caudal fate of the generated oligodendrocytes. Second, the assembly of NECs/NPCs into glial spheres promotes the enrichment of the progenitors committing to oligodendrocyte lineage. While this glial sphere aggregation step has been present in a vast variety of established protocols [16,30,35], Fossatti’s group [38] elucidated the mechanism leading to the enrichment of the glial progenitors via the generation of the glial spheres. In their experimental system [38], they introduced a GFP reporter to trace the expression of OLIG2. The observations uncovered that only GFP^+^ cells formed glial spheres, while GFP-negative cells were not incorporated into these aggregates/glial spheres. These findings further provide evidence for the enrichment of OLIG2^+^ cells during aggregate formation. It has been shown that OLIG2-expressing progenitors inhibit neuronal transcription factors such as PAX6 and neurogenin 2 (NG2) and enhance the expression of transcription factors promoting oligodendroglial fate such as NKX2.2 and SOX10 [7,13,39]. Third, glial media containing small molecules and growth factors promoting oligodendrocyte survival (e.g., biotin, cAMP), development (e.g., PDGF-AA and IGF), and maturation (e.g., T3) has been extensively used to expand, sustain, and mature oligodendrocytes in vitro. While the composition of the glial media might vary slightly between protocols, we will refer to the media containing the molecules mentioned above as glial media in the next sections of the manuscript.

While these ESCs-based protocols provided a foundation for the generation of ESC-derived oligodendrocytes and promised an avenue for regenerative therapy, they did not address the potential rejection of the donor cells. This issue can become especially acute in demyelinating disorders associated with the activation of the immune system, such as multiple sclerosis (MS). Therefore, the generation of oligodendrocytes derived from iPSC lines was pioneered by Goldman’s group [30]. They utilized three different iPSC lines derived from three individuals generated through distinct reprogramming protocols and established a protocol for OPC differentiation in six steps with a culture duration of up to 150 days. The incorporation of the critical steps from the previous protocols, such as the generation of EBs, followed by the ventral patterning through the application of purmorphamine (PM) [40] and the caudal patterning with RA induced the appearance of OLIG2^+^/NKX2.2^+^ double-positive cells, defined as early pre-OPCs. However, in contrast to previous protocols, a more extensive period of time was covered by PM treatment (from day 20 and up to day 35–40), and the transition from early pre-OPCs to the late pre-OPCs following the removal of RA was promoted through the treatment with FGF-2 in the presence of PM. At the final stage, the gliogenic spheres were dissociated and plated in the glial medium mentioned above for extended culture of up to 3–4 months, resulting in a population of OPCs suitable for maturation in vitro as well as for transplantation experiments. Indeed, in vitro terminal differentiation resulted in the appearance of O4^+^ and MBP^+^ oligodendrocytes capable of myelination when co-cultured with human fetal cortical neurons. Furthermore, the transplantation of OPCs into shiverer mice was performed after 4 months in culture, resulting in the production of compact myelin and the robust myelination of shiverer mice forebrain, as well as prolonged life expectancy in these mice [30].

While these groundbreaking works open endless possibilities for regenerative medicine, the extensive length of these protocols, consistency in the applications, and the generation of oligodendrocytes with spinal-cord-like characteristics (based on the utilization of RA) still presented a challenge. Fossatti’s group [38,41] established a robust protocol for iPSC-derived oligodendrocytes that can produce a large number of oligodendrocytes, and is extremely consistent across multiple cell lines [38,41,42]. The key features of the developed protocol included adherent cultures (and not EBs) exposed to dual SMAD inhibition, viz. inhibition of bone morphogenetic protein (BMP) pathways by LDN-193189 (LDN), and transforming growth factor-β (TGF-β) pathways by SB431542 (SB) for the generation of neural stem cells [43]. The treatment with RA was applied from the beginning of differentiation and enhanced the production of OLIG2^+^ progenitors. Importantly, no exogenous FGF was used in this protocol, and the induction of OLIG2^+^ progenitors was achieved through the combination of dual SMAD inhibition with RA and the stimulation of the SHH pathway by a smoothened agonist (SAG) (instead of PM). Through this protocol, the O4^+^ cells were detected as early as day 50, and their numbers increased significantly around day 75; making this protocol dramatically shorter (75 days) compared to the previously established protocol of 120–150 days [30]. Importantly, this protocol has been validated with four human iPSC lines derived from patients with MS, suggesting robustness and consistency. Furthermore, on day 75, fluorescence-activated cell sorting (FACS)-purified O4^+^ oligodendrocytes were engrafted into the forebrain of shiverer mice resulting in oligodendrocyte maturation and compact myelin formation.

While this protocol enabled many labs to reproducibly differentiate iPSC lines into myelinogenic oligodendrocytes, it did not address the effect of RA on the “caudalization” of the generated oligodendrocytes, nor the features that might affect the implementation of the oligodendrocytes with “spinal cord-like characteristics” in the studies targeting myelination processes in the brain. This aspect might be increasingly important for the studies focused on oligodendrocyte dysfunction and white matter pathology in disorders implicated in cognitive disabilities and mental health, such as Down Syndrome (DS) [36], Autism Spectrum Disorders (ASD) [44], and schizophrenia [42].

Thus, this facet has been addressed in the following studies through the manipulation of the existing protocols to generate forebrain iPSC-derived oligodendrocytes [45]. The ventral identity of the NECs/NPCs has been achieved through the activation of SHH pathway by either PM [30,35,45] or SAG [36,43]. However, the assignment of the anterior identity was achieved through the inhibition of the Wnt-mediated signaling pathway aimed to potentiate the anterior/forebrain identity of the NECs/NPCs [46,47,48]. The Wnt signaling pathway has been considered to be one of the main pathways regulating the anterior–posterior axis in the developing embryo. The anterior portion of the neural tube expresses molecules antagonizing Wnt expression, and the posterior portion is enriched with the cells expressing Wnt proteins [45,47]. Therefore, Wnt antagonists XAV939 [45,48] or C59 [46] have been used to assign forebrain identity to iPSC-derived oligodendrocytes, as was confirmed through the expression of the forebrain marker FOXG1. However, no comparison was conducted in these studies to evaluate the differences between the NECs/NPCs and OPCs generated in the presence of the RA versus those generated through exposure to the Wnt antagonists.

In our recent work [36], we generated iPSC-derived oligodendrocytes from two female isogenic lines derived from individuals with DS. As a part of our study, we have assessed the effect of the trisomy on the generation of transitioning NPCs (tNPCs) that adopt ventral and glial fate in the context of the regional patterning induced by the treatment with RA (caudal properties) versus Wnt inhibitor-forebrain characteristics. One of the main findings of our study showed that the trisomy 21-related dysregulation in the SHH signaling pathway is prominent in the forebrain-like tNPCs, but not in the caudal spinal cord-like tNPCs. Additionally, the differential expression of thousands of other genes between trisomic and diploid tNPCs was triggered by patterning-specific conditions (forebrain tNPCs vs. spinal cord tNPCs), highlighting the impact of the spatial heterogeneity that should be accounted for through the modifications of the signaling pathways defining regional/spatial identity.

The generation of oligodendrocytes from stem cells in 2D models tremendously expanded our knowledge regarding different aspects of oligodendrocyte biology in health and disease, which includes epigenetic changes governing cellular development, heterogeneity, and functional genetics, among others [49,50,51]. For a more detailed and comprehensive survey of the differentiation of oligodendrocytes in 2D culture using additional approaches, we would refer readers to a few recently published reviews [52,53].

Important knowledge gained from these studies empowered the researchers to push the technological boundaries and cultivate oligodendrocytes in the three-dimensional (3D) spheroid/organoid system. This new system overcomes the caveats of the 2D models such as the limited viability of mature oligodendrocytes in the absence of neuronal axons or using synthetic material as a substrate for myelination. In addition, 2D oligodendrocyte systems do not allow spatial and temporal co-development with other glial and neuronal cells mimicking the intercellular interactions, cytoarchitecture, and cellular milieu that oligodendrocytes encounter in vivo.

## 3. Oligodendrocytes and Myelin in iPSC-Derived Organoids

Since their establishment by Knoblich’s group [54], human 3D cerebral organoids have become a valuable tool for the investigation of diverse cellular mechanisms in a physiologically relevant cellular environment, preserving major cell types and the crosstalk between cell lineages. The presence of OPCs and mature oligodendrocytes in cerebral organoids was initially questionable until the detection of OPCs expressing OLIG1 and O4, but not MBP-positive cells in 5-month-old cerebral organoids [55]. Culturing ESC-derived cerebral organoids extensively for 6–8 months resulted in the production of more mature oligodendrocytes expressing MBP [56]. Additional transcriptomic studies in the organoids supplied evidence for the presence of oligodendrocytes in different types of organoids [57,58]. A recent study [59] focused on the organization of the dopaminergic neurons in the midbrain organoids reported that oligodendrocytes can develop, mature, and even myelinate within the midbrain organoids. This suggests that even in the absence of exogenously added morphogens and growth factors explicitly promoting the specification and maturation of oligodendrocytes, human brain organoids are capable of some generation of oligodendrocytes that might reach a certain level of maturity [56,60].

However, elucidation of the processes regulating oligodendrocyte differentiation and myelination in physiological and disease-related conditions requires a more rigorous and consistent cellular system. Tesar’s group [61] developed a reproducible protocol to generate self-organizing oligo-cortical spheroids enriched with OPCs and myelinating oligodendrocytes modifying the organoid protocol originally developed by Pasca’s group [62]. Following dual SMAD inhibition, the induction of OPCs was achieved through the expansion of the innate small population of OPCs present in organoids through the application of the growth factors PDGF-AA, IGF-1, and T3. This protocol primarily recapitulated the dorsal telencephalon-derived oligodendrocytes (third wave of oligodendrocyte generation) [23], which resulted in the generation of mature oligodendrocytes by 20 weeks of differentiation, and the appearance of compact myelin by 30 weeks (see Figure 1).

The researchers also showed the utility of this protocol for the differentiation of the lines derived from patients with congenital leukodystrophy, Pelizaeus–Merzbacher disease (PMD) (see Table 1), and assessed the therapeutic efficacy of promyelinating drugs such as ketoconazole and clemastine.

This advancement was complemented by the additional study conducted by Pasca’s lab [63], which utilized seven human iPSC lines. In contrast to the Tesars group’s protocol [61], oligodendrocyte enrichment was induced through the assignment of the ventral fate to the developing spheroids via activation of the SHH pathway and the inhibition of Wnt signaling, by SAG and IWP-2, respectively. This mimics the embryonic developmental origin of oligodendrocytes arising from the medial ganglionic eminence, and corresponding to the first wave in oligodendrocyte development [23]. The maturation steps were achieved through culturing of the spheroids in a glial medium (see Figure 1). Importantly, the transcriptomic signature of the oligodendrocytes matched that of the primary oligodendrocytes derived from the human cortex indicating the fidelity of oligodendrocyte gene expression patterns through the transition across developmental stages. Finally, oligodendrocytes generated through this protocol exhibited morphological, physiological, and functional properties, and the lamellae of compact myelin surrounding axons were detected at 5 months.

These two fundamental protocols promoted the enrichment and development of oligodendrocytes in the forebrain organoids/spheroids. They paved the way for additional studies that aid in dissecting the inter-cellular interactions and molecular mechanisms mediating oligodendrocyte fate acquisition, migration, differentiation, and maturation in the healthy brain and disorders impacting myelination.

Another significant study that came out around the same time [64], developed region-specific forebrain organoids with dorsal and ventral identity (DFOs and VFOs, respectively) using the previously established OLIG2-GFP reporter line. Following the generation of the organoids through EB and neural rosette selections, 3-week-old organoids were treated for 2 weeks with SHH and PM for the induction of ventral identity in VFOs, or with cyclopamine (CycA) for the induction of the dorsal identity in DFOs (see Figure 1). This approach conceptually represents the first and third waves of oligodendrocyte development described previously in the mouse brain [23]. While both VFOs and DFOs were enriched with oligodendrocytes by week 12 of differentiation, VFOs exhibited OPCs markers as early as 5 weeks that increased drastically by the 12-week time-point, while DFOs showed strong enrichment in OPCs only following treatment with T3 (weeks 9–12). It was suggested that the induction of neuronal maturation in both DFOs and VFOs with BrainPhys^TM^ medium between weeks 7–9 of differentiation enhanced oligodendrocytes maturation in the VFOs, but not in the DFOs, as significantly increased levels of MBP were detected in VFOs and no MBP expression was observed in the DFOs. The differences can be also attributed to the fact that DFOs were cultured in Neuronal differentiation media (see Figure 1) during the early stages of the protocol (weeks 5–7), while OPC-promoting/glial media was added to VFOs during this period, potentially impacting the numbers and maturation capacity of the oligodendrocytes in the DFOs. The lack of VFO exposure to the OPC-promoting/glial media in Kim’s protocol [64] can also explain the discrepancy in oligodendrocyte differentiation capacity in the “non-ventralized” spheroids in Madhavan’s protocol. In that study [61], a small population of OPCs was expanded early in the development by media supplementation with PDGF-AA and IGF-1 (prior to the administration of T3), while in Kim’s protocol [64], DFOs were maintained in a neuronal differentiation media medium (see Figure 1) during early stages of the protocol. Remarkably, this study also generated fused forebrain organoids (FFOs) by assembling week 9 DFOs with week 5 VFOs. This resulted in (i) a massive migration of OLIG2 cells from the VFOs to DFOs during the first 3 weeks after fusion; (ii) significantly increased oligodendrocyte maturation as compared to both VFOs and DFOs 6 weeks after fusion, and (iii) the appearance of loosely packed myelin structures at week 15. The authors also concluded that at this advanced stage of maturation, VFO-derived oligodendrocytes were outnumbered by DFO-derived oligodendrocytes, ultimately recapitulating the studies in mouse models [23]. There are several questions that can be answered using this model: Which oligodendrocyte population is capable of myelin sheath generation within the FFOs? How is migration capacity of oligodendrocytes from the ventral to dorsal domains affected in disease models? It is also intriguing to know what the implications on the populations of neurons and oligodendrocytes in VFOs and DFOs would be had they been cultured in the same oligodendrocyte media in the early stages of the protocol.

Despite these advantageous pioneering studies describing the generation of organoids containing myelinating oligodendrocytes, the representation of myelin is limited. Moreover, the axonal ensheathment is only partially compacted, and largely accompanied by a lack of paranodal and nodal structures. James et al., 2021, suggested the regional patterning towards forebrain oligodendrocytes may lay at the root of insufficient myelination as human cortical myelination takes place late in development, while the acquisition of myelin in the spinal cord happens significantly earlier [65]. This hypothesis is supported by the presence of nodes of Ranvier-like structures in the midbrain organoids [59]. Thus, spinal cord-patterned organoids were enriched with mature myelinating oligodendrocytes through early-stage exposure to RA and SAG to promote caudal and ventral patterning, respectively, and subsequent culturing in the glial media containing PDGF-AA, IGF-1, and T3 to promote oligodendrocyte development [65]. These spinal cord organoids were then cultured at the air–liquid interface to further enhance the metabolic support to oligodendrocytes (and other cell types) within the organoids, leading to the formation of compact myelin, and the detection of nodal and paranodal structures. In line with other works [36,64], this study highlighted the significance of regional patterning to oligodendrocyte biology, as well as the utility of the protocol to disease modeling by showing abnormal paranode assembly using iPSC lines derived from patients with a mutation in *Neurofascin-155* (Nfasc155) (see Table 1).

In an attempt to accelerate the timeline required for the generation of mature oligodendrocytes in organoids, Wolvetang’s group [66] developed a shorter protocol of 42 days (as compared to 210 [61], 100 [63], and 105 [64] days in the previous studies) required for the formation of myelin ensheathment in the organoids. A human iPSC SOX10 reporter line was utilized to monitor the appearance and maturation of oligodendrocytes. Application of dual SMAD inhibition for 3 days and FGF2 for 4 days, followed by the exposure to a single combination/cocktail of small molecules led to the appearance of the SOX10 positive cells on day 14 and a subsequent robust expression of O4 and MBP at day 42. Similarly to the earlier work [61], despite the detection of concentric myelin layers, only a limited number of axons were myelinated. In addition, while cortical neurons and astrocytes were also developed under this protocol, it remains to be established how this oligodendrocyte-promoting intensive protocol impacts neurogenesis and astrogenesis, neuronal maturation, and neuronal electrophysiological properties.

An additional study [67] was carried out to minimize the timing required for OPCs in 2D and 3D cultures. To achieve this goal, they overexpressed two transcription factors SOX10 and OLIG2 through lentiviral transduction. P2A sequence was introduced into each one of the constructs to trace the expression of the overexpressed SOX10 with mCherry and OLIG2 with EFGP. Forebrain oligodendrocyte-enriched spheroids were generated following the exposure of the hESC and iPS-derived EBs to dual SMAD inhibition, and the ventral patterning of the spheroids was induced through the application of SAG. These ventrally patterned spheroids were transduced with LV-SOX10 -mCherry and LV-OLIG2-EGFP or day 13 post-differentiation and kept in the glial-induction medium for 10 days to support the generation and expansion of the NPCs committing to oligodendrocyte lineage in the forebrain spheroids. The media containing molecules and growth factors promoting oligodendrocyte differentiation and maturation were applied between days 23 and 60 (see Figure 1). The overexpression of SOX10 and OLIG2 potentiated the enrichment with differentiated oligodendrocytes reflected in the detection of O1, O4, and MBP-positive cells in the spheroids on day 60 accompanied by the formation of myelin lamellae around axons detected at the edge areas. Furthermore, the transplantation of the 40-day oligodendrocyte-enriched spheroids into newborn SCID mice resulted in the detection of myelinated axonal segments originating from human cells, and concentric myelin sheaths at 12 weeks post-transplantation. Interestingly, the findings in the 2D system in this study showed that the overexpression of SOX10 or OLIG2 alone is not sufficient to produce OPCs expressing O4, and the overexpression of both transcription factors is required. On one hand, this contradicts the previous study, in which the expression of SOX10 was shown to be sufficient to drive the cells of oligodendrocyte lineage to maturation [72,73]. On the other hand, the study by Xiong’s group [67] is in line with the requirements for the three transcription factors: SOX10, OLIG2, and NKX6.2 (SON) to convert human iPSCs into myelinating oligodendrocytes [74]. Moreover, SON-mediated conversion into oligodendrocytes is more robust than the one induced by SOX10 alone [75]. In addition, the SON-based protocol has been shown to efficiently generate oligodendrocytes from fibroblasts (direct conversion), while preserving epigenetic age-related signatures [76]. While the transcription factors-driven approach provides an opportunity for an accelerated generation of the spheroids enriched with oligodendrocytes capable of myelination in vivo and in vitro, the transcription factor overexpression-mediated induction of oligodendrocytes precludes the investigators from the assessment of the intrinsic forces driving oligodendrocyte development in healthy and perturbed conditions.

Monitoring oligodendrocyte development and maturation within a variety of 2D and organoid models appears to be an informative and attractive strategy. As such, PDGFRα [51,77], proteolipidprotein1 (PLP1), and MBP [77] reporter lines were used to monitor the heterogeneity, developmental trajectory, and maturation capacity of iPSC-derived oligodendrocytes in 2D culture. PLP1 tagged with super-fold GFP (PLP1-sfGFP) reporter lines were generated to trace oligodendrocyte biology in an additional organoid model called the “human brain microphysiological system” (bMPS) [68]. bMPS contains mature glutamatergic, dopaminergic, and GABAergic neurons as well as astrocytes and oligodendrocytes with more than 40% of myelinated axons [78]. The protocol is based on the generation of small sized (<300 µm) spheroids aggregated from iPSC-derived NPCs that are maintained in culture for about 8 weeks on a gyratory shaker. Remarkably, the application of the differentiation medium (see Figure 1) resulted in the detection of OLIG2 and MBP-positive cells as early as 2 weeks post-induction, and the appearance of myelin sheaths at 8 weeks. Three important features distinguish this protocol from the others: (i) the very small size of the spheroids potentially preventing the generation of a necrotic core and providing improved oxygenation for different cell populations, including oligodendrocytes, (ii) the absence of the classical “glial medium” promoting oligodendrocyte development, and (iii) the presence of mature, electrophysiologically active neurons. Based on this and previous reports [64,65], one might suggest that the signal originating from mature neurons displaying spontaneous electrical activity, sufficient metabolic support, and oxygenation might be the driving forces for the induction of oligodendrocyte maturation. Importantly, PLP1-sfGFP reporter aided in the tracing of the PLP signal colocalized to MBP along the neurofilaments without interrupting the functionality of PLP [68]. Noticeably and unexpectedly, the exposure of bMPS to cuprizone enhanced the expression of PLP1-sfGFP signal concomitant with the reduction in mature oligodendrocytes positive for both PLP and MBP, exposing the potential compensatory mechanism activated during demyelination. Thus, the utilization of fluorescent reporter tags can facilitate the mechanistic understanding of the developmental and maturation stages of oligodendrocytes during physiological and pathophysiological conditions.

In their most recent study [69], the same group further optimized the bMPS protocol to promote the expansion of astrocyte and oligodendrocyte populations through the enrichment of the previously described differentiation medium [68] with the components of the glial medium (weeks 2 to 8) and signaling molecules essential for the terminal differentiation media (weeks 8–15, see Figure 1). This optimization resulted in an expansion of astrocytes and oligodendrocytes as was evident from the elevated levels of O4, MBP, and GFAP as well as enhanced myelin ensheathment of axons. Remarkably, boosting oligodendrocyte and astrocyte maturation did not compromise neuronal maturation and activity. On the contrary, glial migration and outgrowth of neurons, as well as calcium signaling, were enhanced under the effect of the glial medium and were more prominent at 10 weeks following differentiation. This suggests that the interactions between neurons and glial cells in a more mature environment may lead to an enhanced functional maturity of the lineages and improved cellular circuitry within the organoid system.

Another recent study by Shi’s group [70] provided an opportunity for a more detailed evaluation of myelin ultrastructure within the organoids through modifications to the previous protocol developed by Fossati’s group [41]. The initial workflow was designed for the generation of oligodendrocytes in 2D culture and included the plating of the glial spheres for 4 weeks for the expansion of the OPCs. But Shi’s group continued the culturing of the spheres/oligodendroglial spheroids generated from 3 iPSC lines for up to 18 weeks in the glial media without plating (see Figure 1). Assessment of oligodendrocyte maturation and myelin ensheathment was performed every 2 weeks. OLIG2^+^/MBP^+^ cells were detected within the organoids starting at week 8, and their number gradually increased to week 12, accompanied by the morphological changes associated with extensive branching. The first signs of the axonal ensheathment were observed at 12 weeks followed by the increase in total myelin length and the frequency of myelin sheaths colocalized to neurofilaments between weeks 12 to 20. The expression of Caspr, a paranodal marker, was sporadically observed at 18 weeks; however, by week 30, its expression colocalized consistently and exclusively to the paranodal regions. This indicates the formation of the nodes of Ranvier and proper myelination, which was further supported through the examination by EM. This protocol has been applied to iPSC lines derived from a patient with Canavan disease (CD), a type of leukodystrophy that leads to demyelination related to abnormal N-acetyl- aspartate (NAA) metabolism (see Table 1). Treatment of the CD spheroids with NAA was sufficient to induce a disruption in myelin sheaths, suggesting a high relevance of this model for studying demyelination disorders, and exploration of potential therapeutics. While this protocol resulted in the generation of neurons, astrocytes, and oligodendrocytes, this work did not investigate neuronal heterogeneity within the spheroids, nor did it assess the regional patterning of the generated spheroids. This is important since the RA and SAG have been used at the beginning of differentiation, potentially assigning a ventral–caudal identity to the generated spheroids.

While the former protocol utilized and expanded oligodendroglial spheroids, specifically enriched with maturing oligodendrocytes to study CD, another recently published work [71] used the original protocol for the generation of cerebral organoids [54] to elucidate the cellular mechanisms taking place in different variants of MS (see Table 1). The investigators identified aberrations in cell cycle progression and defective proliferation of NPCs that resulted in excessive neurogenesis, and diminished oligodendrogenesis in iPSC MS lines. Noticeably, no mature oligodendrocytes were detected using this model, preventing further investigation of oligodendrocyte maturation. Remarkably, the specific pathological features related to this demyelinating disease mechanism were detected in an organoid model with no purposeful enrichment in oligodendrocyte lineage and promotion of oligodendrocyte maturation. The last two mentioned studies provide evidence that versatile organoid models may be applied to study oligodendrocyte lineage commitment, differentiation, maturation, and myelin production across development to uncover stage-specific events in oligodendrocyte biology in a vast variety of conditions.

## 4. Conclusions

A variety of iPSC-derived organoid models for the induction of oligodendrocyte progenitors and mature myelinating oligodendrocytes have been established and optimized in recent years. Oligodendrocyte-containing 3D systems provide a new versatile platform for longitudinal exploration of phenotypical and functional oligodendrocyte properties, active myelination of the surrounding neurons, and interactions between neurons and glial cells. Furthermore, regionally patterned oligodendrocyte-containing organoids enable us to inquire into oligodendrocyte biology with preserved regional characteristics in healthy and perturbed brain development as well as in myelin-related and neurodegeneration-related disorders. However, some critical questions remain unanswered.

First, studies in zebrafish showed regional differences in the oligodendrocyte abundance and the number of ensheathments produced by a single oligodendrocyte between ventrally and dorsally derived populations [21]. It would be informative to know whether such differences persist in human iPSC-derived oligodendrocytes present in organoids with a distinct regional identity.

Second, a significant number of the protocols aiming to enrich/expand the oligodendrocyte population within the organoids utilize the supplementation with glial medium (e.g., PDGF-AA, IGF, T3). No study so far has shown how these manipulations and the timing of these interventions affect neuronal activity, and whether they inhibit neuronal maturation, or if the presence of mature oligodendrocytes itself promotes neuronal activity.

Third, while many studies report the presence of concentric myelin layers, the extent of myelin compaction possible within the organoids is not entirely clear. Moreover, the methods for reliable and consistent quantification of compact myelin lamellae are yet to be defined and standardized.

Fourth, microglia, the resident immune cells of the CNS, originate from mesoderm, and thus are not naturally generated in the neuroectoderm-centric organoids. However, recent studies have assessed the effect of exogenously incorporated microglia on neuronal activity and synaptogenesis [79,80]. Although microglia provide trophic support to oligodendrocytes to promote their differentiation, the impact of microglia on oligodendrocyte biology within the organoids has not been addressed.

Finally, several recent studies evaluating the effect of the xenotransplantation of human organoids into rodent brains reported an extensive maturation of neuronal networks accompanied by the presence of oligodendrocyte population within the xerographs, but not within the age-matched control organoids [81,82]. The cells expressed OPC-associated signature with no expression of the markers of mature oligodendrocytes. More extensive studies are required to understand the impact of the maturation-promoting in vivo environment on the cells of the oligodendrocyte lineage.

## Figures and Tables

**Figure 1 cells-13-00674-f001:**
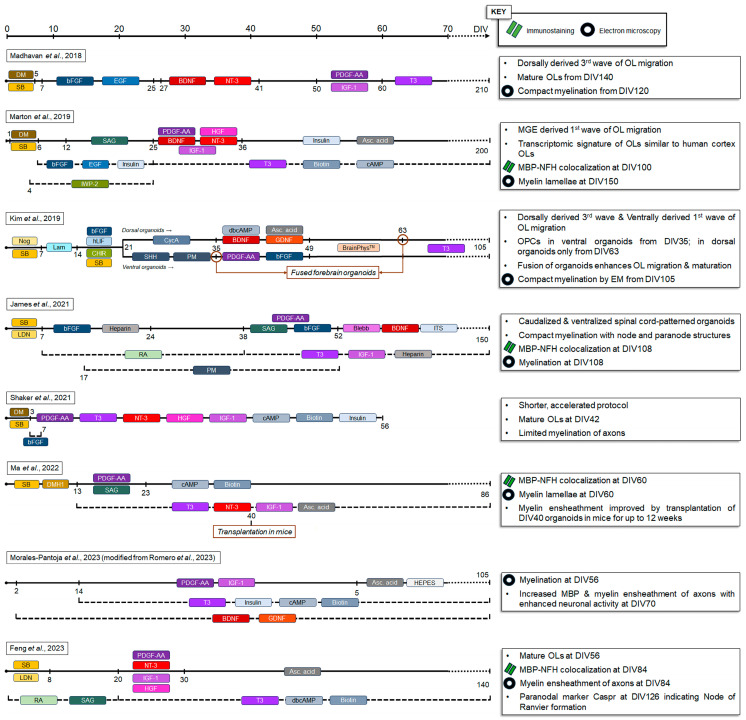
Schematic overview of various 3D organoid protocols utilizing small molecules to promote oligodendrocyte generation, differentiation, and maturation. Highlights of each protocol including the earliest timepoint at which myelination was confirmed are depicted to the right of each protocol’s timeline along with the experimental technique used for the assessment, such as colocalization of MBP with NFH by immunostaining, and myelin layer number and thickness by electron microscopy [61,63,64,65,66,67,68,69,70]. (Asc. acid: Ascorbic acid; BDNF: Brain-derived neurotrophic factor; bFGF: Basic fibroblast growth factor; Blebb: Blebbistatin; cAMP: Cyclic adenosine monophosphate; CHIR: CHIR99021; CycA: Cyclopamine; dbcAMP: Dibutyryl cyclic adenosine monophosphate; DIV: Days in vitro; DM: Dorsomorphin; DMH1: Dorsomorphin homolog 1; EGF: Epidermal growth factor; GDNF: Glial cell line-derived neurotrophic factor; HEPES: 4-(2-hydroxyethyl)-1-piperazineethanesulfonic acid; HGF: Hepatocyte growth factor; hLIF: Human leukemia inhibitory factor; IGF-1: Insulin-like growth factor-1; ITS: Insulin-Transferrin-Selenium; IWP-2: Inhibitor of Wnt production-2; Lam: Laminin; LDN: LDN-193189; MAP2: Microtubule-associated protein 2; MBP: Myelin basic protein; MGE: Medial ganglionic eminence; NFH: Neurofilament heavy chain; Nog: Noggin; NT-3: Neurotrophin-3; OL(s): Oligodendrocyte(s); PDGF-AA: Platelet-derived growth factor homodimer subunit AA; PM: Purmorphamine; RA: Retinoic acid; SAG: Smoothened agonist; SB: SB-431542; SHH: Sonic hedgehog; T3: Triiodothyronine).

**Table 1 cells-13-00674-t001:** 3D organoid models used to study myelin-related diseases.

Study	Disease	Key Finding(s)	Validation Method(s)
Madhavan et al., 2018 [61]	Pelizaeus–Merzbacher disease	▪ Increase in number of MYRF^+^ oligodendrocytes on PLP1 deletion and decrease on PLP1 duplication ▪ Perinuclear retention of PLP1 reversed by treatment with the protein kinase R-like endoplasmic reticulum kinase (PERK) inhibitor, GSK2656157	Immunohistochemistry Electron microscopy
James et al., 2021 [65]	*Nfasc155*^–/–^ demyelination disorder	▪ Lack of paranodal localization of glial NFASC and CASPR indicating disruption in formation of the paranodal axo-glial junction (PNJ)	Immunohistochemistry
Feng et al., 2023 [70]	Canavan disease	▪ Reduction in myelin sheath number and total length ▪ Increased vacuolation in myelin sheath ▪ More number of scattered and swollen myelin sheaths	Immunohistochemistry Electron microscopy
Daviaud et al., 2023 [71]	Multiple sclerosis	▪ Imbalance in neural stem cell proliferation/neurogenesis due to a shift in cleavage plane angle towards asymmetrical division ▪ Reduced expression of the cell cycle inhibitor, p21 ▪ Decrease in OLIG2+ oligodendrocyte population	Immunohistochemistry

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
