# Peer review of "Identity and Maturity of iPSC-Derived Oligodendrocytes in 2D and Organoid Systems"

_cells, 2024, doi:10.3390/cells13080674_

Round 1

Reviewer 1 Report

Comments and Suggestions for Authors

Comments on the review manuscript by Ella Zeldich and Sandeep Rajkumar

Entitled ‘Identity and Maturity of iPSC-Derived Oligodendrocytes in 2D 2

and Organoid Systems‘ submitted to Cells

The review is a timely and concise description of the field and an important contribution, it is generally well written and clearly deserves publication.

Major

The only major concern is that ‘essence’ of what is presented as Fig 1 is somewhat unclear …and hard to catch. I would strongly suggest to split the Figure in Fig. 1 (2D protcols) and Fig. 2 (3D protocols) and to add the citation numbers to better reach out for the corresponding text passage. And, it may be very helpful to have to each of the presented protocols a short statement of why this protocol has this or that advantage and/or reason to be presented as part of the figure. Table 1 deserves a header for each column.

Minor

-First sentence: ‘Oligodendrocytes, the myelinating cells of the central nervous system (CNS), are chiefly responsible for myelin ensheathment of axons, provide trophic support and protection to neurons, and regulate iron homeostasis.’ 

While ensheathment is cleary textbook knowledge, the trophic support and iron homeostatis are not and deserve citations such as: Trophic support PMID: 37703600 and/or PMID: 18558866; PMID: 32531201; PMID: 29078110

-p7 line 280 correct the phrase ‘… requires more a …’ to ‘… requires a more …’

-Regarding the direct conversion of OLs by transcription factors (p10) the following citations are missing and need to be cited:

PMID: 28246330 Conversion of human iPSCs by Olig2, Sox10 and NKX6.2 (SON) supports together with PMID: 35053357 that the SON protocol is more efficient in generating OLs than other protocols.

Moreover, in PMID: 33770499, the SON protovol has been applied to efficiently convert also human fibroblasts into OLs, which is of importance in keeping epigenetic marks alive.

Author Response

Major

The only major concern is that ‘essence’ of what is presented as Fig 1 is somewhat unclear …and hard to catch. I would strongly suggest to split the Figure in Fig. 1 (2D protcols) and Fig. 2 (3D protocols) and to add the citation numbers to better reach out for the corresponding text passage. And, it may be very helpful to have to each of the presented protocols a short statement of why this protocol has this or that advantage and/or reason to be presented as part of the figure. Table 1 deserves a header for each column.

We thank the reviewer for the suggestions and input. We feel the we significantly improved the quality of the manuscript during the revision.

A comment related to our Fig. 1. Since our review article is highlighting primarily the development of oligodendrocytes in iPSC-derived 3D organoid models, we only show an overview of different organoid protocols in Figure 1, while the 2D studies are not presented in the figure. In fact, the 2D models for oligodendrocyte differentiation are discussed in detail in McCaughey-Chapman and Connor, 2023. We understand that the placement of the figure under the 2D section is confusing. Thus, we moved the figure to Section 3 where the protocols displayed in the figure are discussed (after line 291). The figure legend (line 292) has also been corrected slightly to specify that only 3D organoid protocols are displayed in the figure.  

The protocols shown in the figure are the only ones currently available and we present each one of them in detail to highlight the various small molecules used across different approaches, as the manuscript focuses on the regional patterning of oligodendrocytes. In addition, the figure also shows the time points where myelination had been assessed and we improved the design of the figure to better address this. Following the reviewer’s suggestion, we expanded Fig. 1 and incorporated another rubric describing the advantages of each protocol. In addition, each of these protocols is discussed in detail in the main text under Section 3 “Oligodendrocytes and Myelin in iPSC-Derived Organoids” elaborating the advantages and limitations of each protocol.  

The headers have been included in Table 1.

Citation numbers for each protocol have been included in both Figure 1 and Table 1 as the reviewer suggested.

Minor

-First sentence: ‘Oligodendrocytes, the myelinating cells of the central nervous system (CNS), are chiefly responsible for myelin ensheathment of axons, provide trophic support and protection to neurons, and regulate iron homeostasis.’ 

While ensheathment is cleary textbook knowledge, the trophic support and iron homeostatis are not and deserve citations such as: Trophic support PMID: 37703600 and/or PMID: 18558866; PMID: 32531201; PMID: 29078110

We thank the reviewer for bringing this point to our attention and for providing us with the relevant references. These have been included in the main text (p1 line 29)

-p7 line 280 correct the phrase ‘… requires more a …’ to ‘… requires a more …’

This error has been corrected (line 265 now).

-Regarding the direct conversion of OLs by transcription factors (p10) the following citations are missing and need to be cited:

PMID: 28246330 Conversion of human iPSCs by Olig2, Sox10, and NKX6.2 (SON) supports together with PMID: 35053357 that the SON protocol is more efficient in generating OLs than other protocols.

Moreover, in PMID: 33770499, the SON protocol has been applied to efficiently convert also human fibroblasts into OLs, which is of importance in keeping epigenetic marks alive.

The reviewer has rightly pointed out that we missed important citations related to the transcription factors-mediated conversion. These citations are now incorporated. In addition, the direct differentiation of fibroblasts into oligodendrocytes using the SON method has indeed been shown to be more efficient and capable of preserving epigenetic signatures. This oversight has been rectified by including these findings in the main text (lines 423-432) along with the relevant references provided by the reviewer.

Reviewer 2 Report

Comments and Suggestions for Authors

I really appreciated the effort made by the authors to summarise a litterature not easy to resume. The scheme in figure 1 is really interesting as it compares work of many different labs. Text is clear and well discussed.

It will be a good review for all those are starting to deal with iPSC-derived OL and organoids. The issue is hot as the models derived from patients iPSCs are of interest of academic as well as pharma. I support the publication with no changes

Author Response

Thanks for your positive comments

Round 2

Reviewer 1 Report

Comments and Suggestions for Authors

Agree with the edits.